# Application of Positional Entropy to Fast Shannon Entropy Estimation for Samples of Digital Signals

**DOI:** 10.3390/e22101173

**Published:** 2020-10-19

**Authors:** Marcin Cholewa, Bartłomiej Płaczek

**Affiliations:** Institute of Computer Science, University of Silesia, Będzińska 39, 41–205 Sosnowiec, Poland; bartlomiej.placzek@us.edu.pl

**Keywords:** entropy estimation, Shannon entropy, positional entropy

## Abstract

This paper introduces a new method of estimating Shannon entropy. The proposed method can be successfully used for large data samples and enables fast computations to rank the data samples according to their Shannon entropy. Original definitions of positional entropy and integer entropy are discussed in details to explain the theoretical concepts that underpin the proposed approach. Relations between positional entropy, integer entropy and Shannon entropy were demonstrated through computational experiments. The usefulness of the introduced method was experimentally verified for various data samples of different type and size. The experimental results clearly show that the proposed approach can be successfully used for fast entropy estimation. The analysis was also focused on quality of the entropy estimation. Several possible implementations of the proposed method were discussed. The presented algorithms were compared with the existing solutions. It was demonstrated that the algorithms presented in this paper estimate the Shannon entropy faster and more accurately than the state-of-the-art algorithms.

## 1. Introduction

The information entropy is a fundamental concept in computer science [1]. For instance, it is significant in security of operating systems [2,3]. Another well-known application domain of the entropy is information encoding for data compression [4]. Moreover, the Shannon entropy is a useful measure, which allows us to quantify the degree of disorder or chaos [5], evaluate the complexity of compounded systems and the divergence between probability distributions [6]. Apart from the classical entropy concepts, the new and modified entropy definitions are of current research interest. In the related literature, several improved algorithms for entropy evaluation have been developed, such as approximated entropy and sample entropy [7], which are useful in determining the irregularity of biomedical samples [8]. A modified entropy definition has also been used to detect rapid changes in electroencephalography (EEG) signal [9], as well as to recognize neuronal signal peaks [10]. Various modifications of the Shannon entropy concern the search for computationally efficient methods [11]. These estimators include centered Dirichlet mixture (CDM) [10], which can be used for binary vectors. In case of known and finite sample distributions, the Nemenman–Shafe–Bialek (NSB) estimator [12], based on the Bayesian estimator, can be used. Other estimators for finite samples are the James–Stein (JS) [13] and the best upper bound (BUB) estimator [14]. Several implementations of state-of-the-art Shannon entropy estimators are available in the R programming language, including the NSB and JS estimators.

This paper introduces the proposed new entropy definition, which is designed to quickly estimate the Shannon entropy for comparison between data samples. In this method, a relatively small number of operations on data sample is performed to estimate the Shannon entropy. The method is based on a simple mathematical formalism, which is easy for implementation. Such an approach introduces an estimation error; however, it is sufficient for comparison of the entropy between data samples. The application areas of this method include analysis of a large image datasets, biomedical data samples and other, where the fast computations are required. In contrast to the state-of-the-art entropy estimation algorithms, that are dedicated for specific data, the introduced method is general and can be adapted for each type of data. As it was already mentioned, there are several entropy estimation methods, which have been implemented in the R programming language [15]. These implementations are available through the CRAN (Comprehensive R Archive Network) repository. The above estimators are applicable for finite data samples. The entropy library for R language is based on the estimation method presented in [13]. Those algorithms are compared in this paper with the introduced positional entropy approach. Results of this comparison are analyzed in order to confirm the improved performance of the new estimator. It was demonstrated that the proposed algorithm allows us to estimate the entropy in shorter time and with higher accuracy, when compared against previous methods from the CRAN repository.

## 2. Preliminaries

In this section the basic definitions of data sample and entropy are presented. Detailed analysis of the entropy definitions is necessary to understand the relationships between them. As a result of such analysis, it was observed that a correlation exists between the presented entropy definitions. The considered entropies can be expressed in a simple way by taking into account differences between elements of a data sample. Based on this observation, it was postulated that a quick estimation of the Shannon entropy can be performed with use of the proposed positional entropy calculations.

### 2.1. Shannon Entropy

According to the deterministic approach, which is based on set theory, the original definition of Shannon entropy [1] can be considered as a relationship between the cardinality of sets. Such approach is used in this paper to avoid unnecessary formal description that would hinder understanding of the presented ideas. Let *X* denote a non-empty and finite set, which is divided into non-overlapping subsets *X*_1_, *X*_2_, ..., *X_p_* that include every element of *X*. The elements in a given subset *X_i_*(*i* = 1, 2, …, *p*) are interpreted as equivalent or belonging to the same class. Thus, *p* is the number of distinguished classes of the elements in *X*.

**Definition** **1.***Shannon entropy is defined by function*$H_{q}:X\to R$: (1)$$H_{q}\left(X\right)=-\sum_{i=1}^{p}\left(\left(\frac{Card\left(X_{i}\right)}{\sum_{j=1}^{p}Card\left(X_{j}\right)}\right)\log_{q}\left(\frac{Card\left(X_{i}\right)}{\sum_{j=1}^{p}Card\left(X_{j}\right)}\right)\right)$$*where the base q of the logarithm usually takes values 2, e, or 10*.

**Definition** **2.***In order to provide definition of the maximum value of Shannon entropy for set X it is convenient to consider another set Y* = {*Y_i_* : *i* = 1, 2, …, *p*}, *where Card(Y_i_) = max*
_*j* = 1, 2, …, *p*_
*(Card(X_j_)) for each i. Then, the maximum Shannon entropy [16] is defined by function*
$H_{q}^{MAX}:Y\to R$:(2)$$H_{q}^{MAX}\left(Y\right)=-\sum_{i=1}^{p}\left(\left(\frac{Card\left(Y_{i}\right)}{p\cdot Card\left(Y_{1}\right)}\right)\log_{q}\left(\frac{Card\left(Y_{i}\right)}{p\cdot Card\left(Y_{1}\right)}\right)\right)$$

In this paper the Shannon entropy is normalized to interval [0,1] using the following definition:

**Definition** **3.***Normalized Shannon entropy [17] is defined by function H_q_^N^: X, Y*→[0,1]
(3)$$H_{q}^{N}\left(X,Y\right)=H_{q}^{N}\left(X\right)=\frac{H_{q}\left(X\right)}{H_{q}^{MAX}\left(Y\right)}$$*where symbol H_q_^N^(X) is used to simplify the notation*.

### 2.2. Positional Entropy

In this section the new definitions related to positional entropy are presented. The positional entropy of a discrete and finite data sample is a measure, which reflects arrangement of elements in the data sample. Low values of the positional entropy correspond to an ordered arrangement of elements in the sample, while high values reflect chaotic arrangement of the sample elements. Position of the elements in the data sample has a direct impact on the value of positional entropy. Any change of position for a single element of data sample results in a new value of the positional entropy.

The data sample can correspond to a part of digital image, sound signal, readings of a light sensor [18], text (e.g., character string), etc. Examples of commonly used data samples are time series, e.g., the time series of traffic intensity [19], heart rate time series [8,20] or time series of other biomedical signals. In general, a data sample comes from any measured or artificially generated signal. In this paper the data samples are represented by sequences of a finite number of integers. In order to evaluate the entropy of a data sample, the analyzed data sample is divided into a finite number of pairs, as described in the following definition. 

**Definition** **4.***Data sample is a set of data or a set of observations, which can be collected by an individual, a group of persons, a computer software or a business process [21]. In this study data sample X is represented by a set of pairs* {{*a, b*}*: a, b*∈*X*}. *The minimum number of pairs must be equal to n*−1 *and the maximum to*
$\left(\begin{array}{c} n\\ 2 \end{array}\right)$*, where n = Card(X). Each element of the data sample belongs to at least one pair*.

It should be noted here that similar data representation is used for other entropy definitions, e.g., for the approximate entropy [22] or the sample entropy [20].

**Definition** **5.***Pair classification function γ:* {*a, b*}→{0, 1} *determines whether the pair is a difference pair. The pair {a, b} is considered as difference pair when a ≠ b. The function γ is defined as:*(4)$$\gamma \left(\left\{a,b\right\}\right)=\{\begin{array}{cc} 1, & \left|a-b\right|\neq 0\\ 0, & \left|a-b\right|=0 \end{array}$$

In this work the data samples with integer elements are considered, however the above definition can be easily generalized to take into account real numbers.

**Definition** **6.***Let d denote the distance between elements of the difference pair* {*a, b*} *in data sample X. Then it is said that the difference pair is a d-adjacent pair. Note that the minimum distance is d* = 1*, and the maximum distance is d = Card(X)*-1. *Examples of d-adjacent pairs are illustrated in Figure 1.*

**Definition** **7.***Positional entropy of data sample X is defined as quotient of the number of d-adjacent difference pairs to the number of all possible d-adjacent pairs. It is assumed that the whole sample X must be divided into a finite number of d-adjacent pairs* {*a, b*}∈*P_d_, where P_d_ is set of all possible d-adjacent pairs. The positional entropy Enp: P_d_*→[0, 1] *is given by the following formula:*
(5)$$Enp\left(P_{d}\right)=\left[\sum_{\{a,b\}\in P_{d}}\gamma \left(\left\{a,b\right\}\right)\right]Card{\left(P_{d}\right)}^{-1}$$

Figure 2 shows a visualization of the positional entropy for an example of a binary sequence of 10 elements. The black squares in this example correspond to ones and the white squares represent zeros. 

The 1-adjacent pairs are taken into account in this example, thus number of all possible pairs is equal to 9.

It should be kept in mind that according to the proposed method, the sample *X* = {*X*_1_, *X*_2_, ..., *X_n_*} is represented by the set of all possible *d*-adjacent pairs *P_d_*. Thus, the notation *Enp*(*X*) can be used instead of *Enp*(*P_d_*).

**Definition** **8.***Cumulative positional entropy Enp*_1*,* 2*, ..., m*_*:* {*P*_1_
$\cup^{}$
*P*_2_
$\cup^{}$
*...*
$\cup^{}$
*P_m_*} *→* [0, 1] *is defined for the union of pair sets P*_1_*, P*_2_*, ..., P_m_, m*∈*N in accordance with the following formula:*(6)$$Enp_{1,2,\ldots ,m}\left(\underset{i=1}{\cup }P_{i}\right)=\left[\sum_{\{a,b\}\in P_{1}\cup P_{2}\cup \ldots \cup P_{m}}\gamma \left(\left\{a,b\right\}\right)\right]Card{\left(P_{1}\cup P_{2}\cup \ldots \cup P_{m}\right)}^{-1}$$

*If m = Card(X)*-1, *then the cumulative positional entropy is determined for all possible pairs and it is denoted as Enp*_1, 2, ..., *m*_*(X)*.

**Definition** **9.**
*Integer positional entropy*
$\bar{Enp}:P_{d}\to N$
*is defined as the number of d-adjacent difference pairs in set P_d_:*
(7)$$\bar{Enp\left(P_{d}\right)}=\sum_{\{a,b\}\in P_{d}}\gamma \left(\left\{a,b\right\}\right)$$


It should be noted that Equation (7) can be obtained from (6) by removing the division operation. The basic relationship between the positional entropy and the integer positional entropy for data sample *X* = {*X*_1_, *X*_2_, ..., *X_n_*} is described by the following formula:(8)$$Enp\left(X\right)=\bar{Enp\left(X\right)}{\left(\sum_{i=1}^{n}Card\left(X_{i}\right)\right)}^{-1}$$

**Definition** **10.***Cumulative integer positional entropy is defined as Enp*_1*,* 2*, ..., m*_: {*P*_1_ ∪ *P*_2_ ∪ *...* ∪ *P_m_*} → *N*:(9)$$\bar{Enp_{1,2,\ldots ,m}\left(\underset{i=1}{\cup }P_{i}\right)}=\sum_{\{a,b\}\in P_{1}\cup P_{2}\cup \ldots \cup P_{m}}\gamma \left(\left\{a,b\right\}\right)$$

In this study the term “integer entropy” is used to refer to the cumulative integer positional entropy, determined for union of sets *P*_1_, *P*_2_, ..., *P_m_*, where *m* = Card(*X*)-1.

## 3. Relationships between Entropies

The Shannon entropy can be expressed by the positional entropy when the *d*-adjacent pairs of sample elements are appropriately determined. The positional entropy was introduced in this study to speed up the estimation of the Shannon entropy by eliminating the time-consuming logarithm calculations. The integer entropy defined in Section 2 can be considered as a discrete version of the Shannon entropy, if Formula (5) or (7) is generalized to all possible *d*-adjacent pairs, i.e., *d* = 1, 2, …, Card(*X*)-1. It means that the positional entropy is a special case of the Shannon entropy. When estimating the positional entropy, it is not necessary to know the elements *X*_1_, *X*_2_, ..., *X*_n_, like in Equation (1). In this case only the differences between sample elements are relevant.

Congruity of entropies is analyzed in this study as follows. Let *h*_1_ and *h*_2_ denote values of the Shannon entropy and *enp*_1_, *enp*_2_ stand for values of the integer entropy for two data samples. Additionally, it is assumed that *h*_1_, *enp*_1_ and *h*_2_, *enp*_2_ are calculated for the first and second sample, respectively. We will say that congruity of Shannon entropy and integer entropy occurs if (*h_1_*⊥
*h_2_*)⇔(*enp_1_*⊥*enp_2_*), where ⊥ denotes order relation (i.e., stands for >, < or =). The operation ⇔ can only be used for samples of the same size. It should be noted that the congruity can be affected by the number of pairs used for evaluation of the positional entropy.

A comparison of the Shannon entropy with positional entropy for 16 data samples is presented in Table 1. In this example, the length of sample *X* is 4 and the number of the all possible sequences is 16. The sample is presented as a string of ones and zeros within the angle brackets <…>. The rows marked with grey and white show the congruity between the Shannon entropy and the positional entropy. If the congruity condition is not satisfied, the entropy value is given in round brackets. For 1-adjacent pairs, in columns denoted by *Enp*_1_*(X)* and $\bar{Enp_{1}\left(X\right)}$, the congruity is not preserved because too few pairs were used. Based on the example in Table 1 it can be observed that when more *d*-adjacent pairs are taken into consideration then the congruity occurs for a larger number of data samples. It can be also noted that increase of *q* leads to decreased difference between values of Shannon entropy and positional entropy. The logarithm base *q* is an important parameter for the conversion between the Shannon entropy and the positional entropy.

Let us assume that *X* contains all possible binary sequences *S_i_* of length *n.* The number of sequences *S_i_* in *X* equals 2*^n^.* The sequences can be divided into *t* subsets such that *X* = {*Z*_0_*, Z*_1_*, …, Z_t_*}, where *t =* 1 *+ n*/2 when *n* is even, or *t* = Ceil(*n*/2) when *n* is odd. Each subset *Z_j_* (*j* = 0, 1, …, *t*) corresponds to equivalence class [*S_i_*] = {*S_i_*
∈*X: f*_1_(*S_i_*) = *j*
*∨ f*_1_(¬*S_i_*) = *j*}, where ¬ denotes the Boolean negation and function *f*_1_(*S_i_*) returns the number of ones in the binary sequence *S_i_*. According to the above definition, subset *Z*_1_ contains the binary sequences with one symbol 1 and the sequences with one symbol 0. *Z*_2_ contains the sequences with two symbols 1 and the sequences with two symbols 0, etc. The subsets *Z_j_* are called the entropy classes. Figure 3 compares values of Shannon and integer entropy for 65 entropy classes. The entropy values presented in Figure 3 were normalized to the interval [0,1].

Differences between the normalized Shannon entropy and the normalized integer entropy are shown in Figure 4. Similar relationships are observed for different values of *q*. For larger number of entropy classes, the shape of this curve does not change significantly.

The best approximation of Shannon entropy is obtained using the positional entropy when the maximum distance between elements in *d*-adjacent pairs is close to the number of entropy classes. For example, if the sample is a binary sequence of length 128, then the number of classes is 65 and the best results are obtained for the cumulative positional entropy when the maximum distance between elements in *d*-adjacent pairs is *d* = 64. The quality of the Shannon entropy approximation for the above mentioned example is analyzed in Figure 5. 

The chart in Figure 5 clearly shows that the best approximation of the Shannon entropy is achieved when the cumulative entropy is calculated for 1, 2, …, 64-adjacent pairs. It can also be noted that for *m* = 32 and *m* = 96, the results differs significantly with those for the Shannon entropy. The larger the value of *m*, the more accurate is the estimation determined by the integer entropy. There is the possibility of converting the integer entropy to the Shannon entropy and vice versa. The conversion formula is a polynomial of grade at most 6 defined on the finite domain [0,1]. The conversion is time-consuming and therefore is not used in the estimation. An important insight for analysis of the relationship between positional entropy and Shannon entropy is given by the following theorem:

**Theorem** **1.***If positional entropy determined for binary data sample X based on* 1-*adjacent pairs (Enp*_1_*(X)) takes the maximum value, then the cumulative positional entropy Enp*_1, 2, 3,...,*m*_*(X) calculated for all d-adjacent pairs is also equal to the maximum value*.

**Proof** When the binary data sample *X* has the maximum positional entropy for 1-adjacent pairs, then all 1-adjacent pairs are the difference pairs. It means that each 1-adjacent pair consists of 0 and 1, thus the ratio of zeros to ones (*µ*) equals 0.5. In this case the number of difference pairs among all possible pairs for *d* = 1, 2, …, *n*, which is given by *n*^2^(*µ*-*µ*^2^), is maximal because the function *n*^2^(*µ*-*µ*^2^) reaches its maximum for *µ* = 0.5. When we have the maximum total number of difference pairs then the cumulative positional entropy Enp_1, 2, 3, ..., *m*_(*X*) calculated for all *d*-adjacent pairs is maximal, which concludes the proof. □

It should be noted here that the binary data samples considered in the above proof have no difference pairs for even values of *d*. However, the total number of difference pairs is maximal because for odd *d* values all pairs are the difference pairs, e.g., in case of sample (1, 0, 1, 0, 1, 0) there is no difference pair for *d* = 2 and *d* = 4, while there are 5 difference pairs for *d* = 1, 3 difference pairs for *d* = 3, and 1 difference pair for *d* = 5. Thus, the total number of difference pairs reaches the maximum of *n*^2^/4 = 9 pairs (*n* = 6).

It is obvious that if *Enp*_1, 2, 3,..., *m*_(*X*) reaches its maximum, then H_q_ is also maximal. The conclusion of Theorem 1 is that when the positional entropy *Enp*_1_(*X*) for data sample *X* is approaching the maximum value, then the estimation of the Shannon entropy by the *Enp*_1,2,3, ...,*d*_(*X*) will require fewer pairs and the execution time of the estimation algorithm can be reduced. For real-world data the sample reach the maximum positional entropy very rarely. However, in many cases the samples have entropy close to the maximum.

## 4. Algorithms

In this section four algorithms are presented that estimate the integer entropy. In order to reduce execution time, the algorithms were defined by using the basic arithmetic instructions. According to the introduced algorithms, the pairs {*x*_1_, *x*_2_} are added to multiset *P_Diff_* if there is a significant difference between their elements. When execution of the algorithm is finished, the cardinality of *P_Diff_* corresponds to the value of integer entropy. Input data samples for these algorithms (denoted by *X*) are sequences of integers. Indexing of the elements in these sequences starts from 0. Algorithm 1 estimates the integer entropy for *d*-adjacent pairs, where d has a predetermined value (*d*∈[1, Card(*X*)- 1]).
**Algorithm 1.** The integer entropy for single *d*-adjacent pairs of *X*.1: **function** INTEGER_ENTROPY_1_(*X*, *d*)2: *l* ← CARDINALITY(*X*)3: **for** i ← 0 **to** l-*d*
**do**4: *x*_1_ ← *X*[*i*]5: *x*_2_ ← *X*[*i*+*d*]6: **if**
*x*_1_
≠
*x*_2_
**then**7: *P_Diff_* ← *P_Diff_*$\;\cup^{}$ {*x*_1,_
*x*_2_}8: **return** CARDINALITY(*P_Diff_*)

Algorithm 2 evaluates the integer entropy by taking into account *d*-adjacent pairs, where d takes s different values randomly selected from interval [1, Card(*X*)-1]. It should be noted that Algorithm 2 uses Algorithm 1 (function INTEGER_ENTROPY_1_).
**Algorithm 2.** The integer entropy for *s d*-adjacent pairs of *X*.1: **function** INTEGER_ENTROPY_2_(*X*, *s*)2: *l* ← CARDINALITY(*X*)3: **for**
*i* ← 1 **to**
*s*
**do**4: *d* ← RAND_UNIQUE(1, *l*-1)5: *P_i_* ← INTEGER_ENTROPY_1_(*X*, *d*)6: *P_Diff_* ← *P_Diff_*
$\cup^{}$ P_i_7: **return** CARDINALITY(*P_Diff_*)

Parameter s in Algorithm 2 denotes the number of different distances between elements of the analyzed pairs. The maximum value of s is equal to *m* = Card(*X*)-1, as defined for the cumulative integer entropy in Equation (9). It should be noted that for high values of *s* Algorithm 2 analyses more pairs and the computational time is longer. In case of *s* < *m*, the distances between pair elements are chosen randomly to decrease the estimation error. The concept of using the *d*-adjacent pairs for selected values of distance *d* is motivated by the conclusions of Theorem 1. Function RAND_UNIQUE(*a*, *b*) returns a unique integer index from interval [*a*, *b*]. Uniqueness of the randomly generated indices is preserved when the number of calls to this function is less than or equal to *b*-*a*. In order to improve the speed of the algorithm, the method uses a predetermined array of random numbers for each call.

The maximum number of pairs analyzed by Algorithms 1 and 2 is $\left(\begin{array}{c} n\\ 2 \end{array}\right)=\frac{n!}{2\left(n-2\right)!}=\sum_{i=1}^{n-1}i$. In Algorithm 3 the analyzed pairs are selected according to the principle presented in Figure 6, where arrows represent the selected pairs. In this case the number of analyzed pairs is reduced by taking into account every *k*-th element of the sequence. The example in Figure 6 assumes that a sequence of 5 elements is analysed and the parameter *k* equals 1. In this case the number of pairs is 10. The number of analysed pairs decreases when increasing the value of parameter *k*.
**Algorithm 3.** The integer entropy for *d*-adjacent pairs of every *k*-th element in sequence.1: **function** INTEGER_ENTROPY_3_(*X*)2: **for**
*i*←0 **to** CARDINALITY(*X*)-1 **step**
*k*
**do**3: **for**
*j*←*i*+1 **to** CARDINALITY(*X*)-1 **step**
*k*
**do**4: *x*_1_ ← *X*[*i*]5: *x*_2_ ← *X*[*j*]6: **if**
*x*_1_
≠
*x*_2_
**then**7: *P_Diff_* ← *P_Diff_*
$\cup^{}$ {*x*_1,_
*x*_2_}8: **return** CARDINALITY(*P_Diff_*)

For further reduction of the computational time, the Monte Carlo approach [23] was used in Algorithm 4. In this algorithm the function RAND_UNIQUE is used to select elements for each analysed pair. The number of analysed pairs is controlled by parameter *α*, which takes integer values. For *α* = 1 the number of analysed pairs equals Card(*X*)-1. The number of pairs verified by Algorithm 4 increases with the value of parameter α.
**Algorithm 4.** The integer entropy for random *d*-adjacent pairs of *X*.1: **function** INTEGER_ENTROPY_4_(*X*, *α*)2: *l* ← (CARDINALITY(*X*)-1)∙*α*3: *k* ← CARDINALITY(*X*)-14: for *i*←0 to *l*-1 do5: *j*_1_ ← RAND_UNIQUE(0, *k*)6: *j*_2_ ← RAND_UNIQUE(0, *k*)7: *x*_1_ ← *X*[*j*_1_]8: *x*_2_ ← *X*[*j*_2_]9: **if**
*x*_1_
≠
*x*_2_ then10: *P_Diff_* ← *P_Diff_*
$\cup^{}$ {*x*_1,_
*x*_2_}11: **return** CARDINALITY(*P_Diff_*)

The algorithms presented above use different strategies to reduce the number of the analyzed pairs, which has a significant impact on the execution time and on the accuracy of entropy estimation. Theoretical analysis of performance for the proposed algorithms was conducted by calculating their time complexity. In this analysis the big O notation was used. It was assumed that the elementary operation corresponds to checking if a selected pair is the difference pair. Results of this performance analysis are presented by the following inequality:(10)$$O_{\min}\left(n\right)\leq O_{4}\left(\left(n-1\right)\alpha \right)\leq O_{3}\left(\frac{n\left(n-1\right)}{2k}\right)\leq O_{2}\left(\sum_{i=1}^{m}i\right)\leq O_{\max}\left(\frac{n\left(n-1\right)}{2}\right)$$
where O_{2,3,4}_ are the time complexities for Algorithms {2, 3, 4} and O_{min, max}_ are boundary time complexities. The time complexity is calculated for *n*-1 pairs needed to estimate the integer entropy.

The presented algorithms were implemented in R using simple time-efficient instructions without much memory consumption. The parameters *m*, *k* and *α* were chosen experimentally, to minimize the execution time and to achieve highly accurate results of entropy estimation. The analysis discussed in [24] revealed that the optimal value of *α* is 4 for sequences of 32-element alphabet. The average 1-adjacent positional entropy for different sample types is shown in Table 2. These results were obtained for samples of length from 64 to 16384. It was observed that a lower value of *α* can be used if the average *Enp*_1_(*X*) is close to 1. Hence, higher *α* and more pairs have to be used for binary sequences than for text strings to correctly estimate the Shannon entropy. The values of parameters *m* and *k* are presented in Table 3.

## 5. Experiments and Discussions

Usefulness of the proposed method was experimentally verified. Computational experiments were conducted with use of data samples represented by sequences of different length. The sequences were generated by the linear congruent generator of pseudo-random numbers (LCG) [25]. The seed for this generator was selected based on noise from sound and light signals and temperature readings. Initial experiments showed that Algorithm 2 allows us to achieve the congruity of 100%. This result was obtained for alphabet lengths of 2, 8, 32, 256 and sequence lengths of 2^8^, 2^9^, 2^10^, 2^14^. However, the entropy estimation with use of all possible *d*-adjacent pairs is not computationally efficient. Therefore, further experiments were performed to examine the possible acceleration of entropy estimation with use of Algorithms 3 and 4.

The proposed entropy estimation algorithms were implemented in R and compared with algorithms from the CRAN package [15]. The R scripts used for these experiments are available in the GitHub repository (https://github.com/mciment/enp). During experiments the normalized Shannon entropy was estimated using seven compared algorithms for each considered sample. Quality of the results was analysed by taking into account congruity between the estimated entropy values and the exact entropy values calculated in accordance with Equation 3. Values of the congruity metrics were determined as *C*/*A*∙100%, where *C* denotes the number of sequence pairs for which the congruity condition (defined in Section 3) is satisfied, and *A* denotes the number of all sequence pairs. Additionally, the estimation error was evaluated as 100 - Congruity (in percent).

During single test the entropy computations were repeated 100 times for data samples (sequences) of 256, 512 and 1024 elements. Time of these computations was measured. Average execution time was determined for 10 tests. It should be noted that the samples were generated earlier, thus the measured execution time does not involve the generation of data samples. The tests were performed on a computer with the iCore7 8750H processor (6-core). The maximum frequency of the core is 4.1 GHz. Only one CPU core was used for the experiments. Thus, multithreaded implementations would improve the results for all algorithms. The execution time did not include the time needed to generate the test sequences. The following algorithms from the CRAN package were examined: Chao, NSB, Shrink and ML. These algorithms are run with the entropy function of R programming language. The entropy function requires the input data sequence to be pre-processed with discretize function. The time needed to perform this pre-processing is included in the measured execution time.

Results of the experiments are summarized in Table 3. A general observation is that the proposed algorithms (Algorithms 2, 3, and 4) were executed faster and achieved a high congruity in comparison with the state-of-the-art algorithms from the CRAN package. The best results were obtained for Algorithm 4, which involves random selection of pairs. In case of all considered lengths of sequences the shortest execution time and the highest level of congruity were achieved by Algorithm 4.

In Figure 7 the dependency between execution time and sequence length is visualized for the proposed algorithm (Algorithm 4) and the state-of-the-art algorithms (Chao, NSB, Shrink, ML). This chart shows that the execution time for the proposed approach is shorter, and what is more important, the execution time for Algorithm 4 increases slower with increasing sequence size than for the remaining algorithms. This observation confirms the advantages of the introduced method regarding the computational effectiveness.

Details of the experimental results for sequences of 256 elements are presented in Figure 8 and Figure 9. These figures show the execution time and estimation error obtained during particular tests. As illustrated in Figure 8, for all 10 tests the execution time of Algorithm 4 was shorter that for the compared algorithm. Moreover, in case of Algorithm 4 the differences of the execution time for particular tests are significantly lower, which means that the results are more stable.

Values of the estimation error calculated for particular tests are shown in Figure 9. When analyzing these results, it can be observed that in majority of cases, the proposed algorithm shows significant improvement in terms of reduced estimation error. The estimation error for the state-of-the-art algorithms varies considerably between tests. In terms of average estimation error Algorithm 4 outperforms the other compared algorithms.

## 6. Conclusions and Further Work

The experiments reported in this paper has confirmed that the new introduced algorithms allow us to speed up entropy estimation. The execution times for the proposed algorithms are clearly better than those of the existing solutions implemented in R language. Additionally, the estimation error is also reduced when using the proposed approach. The shortest estimation time was obtained for Algorithm 4 together with the relatively low estimation error. Moreover, it was also demonstrated that the use of randomization of data pairs selection leads to improved estimation results. For positional entropy, this is an important fact because fewer pairs selected at random have to be analyzed. The novelty of the introduced method is that it offers the possibility to quickly find an accurate estimation of the Shannon entropy with use of simple arithmetic instructions such as incrementation. The accelerated estimation is based on verification if selected pairs in data sample have different elements. It was shown that positional entropy with appropriately selected *d*-adjacent pairs is related to the existing state-of-the-art entropy measures.

Future research directions include detailed theoretical analysis of the entropy estimation accuracy for the proposed algorithms. Moreover, the fact that the best approximation of the Shannon entropy was achieved for *q* = e suggests the existence of other properties that can be useful from the practical and theoretical point of view. These relations require deeper analysis, which will be conducted in the future. The future works will be also devoted to experiments with sample collected from various real world application. This will allow the positional entropy approach to be adapted for measuring data irregularity and outliers detection in specific domains, e.g., in biomedical research. Another interesting topic for future research on the positional entropy is related to the possible modifications of the difference function, which is used to recognition of the difference pairs.

## Figures and Tables

**Figure 1 entropy-22-01173-f001:**
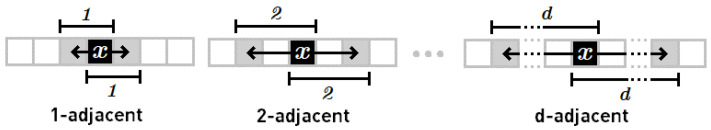
The *d*-adjacent pairs for the middle element of data sample. The white and gray squares represent the elements of the data sample and *x* indicates the middle element.

**Figure 2 entropy-22-01173-f002:**
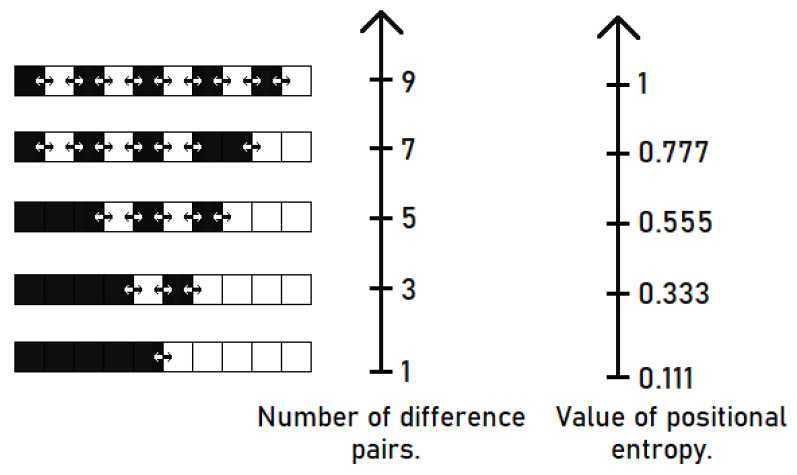
Visualization of positional entropy for binary sequence.

**Figure 3 entropy-22-01173-f003:**
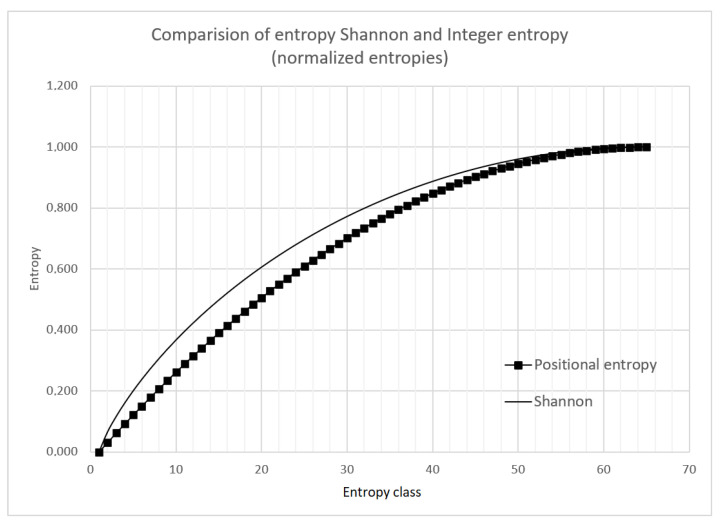
Comparison of Shannon entropy and positional entropy for 65 entropy classes.

**Figure 4 entropy-22-01173-f004:**
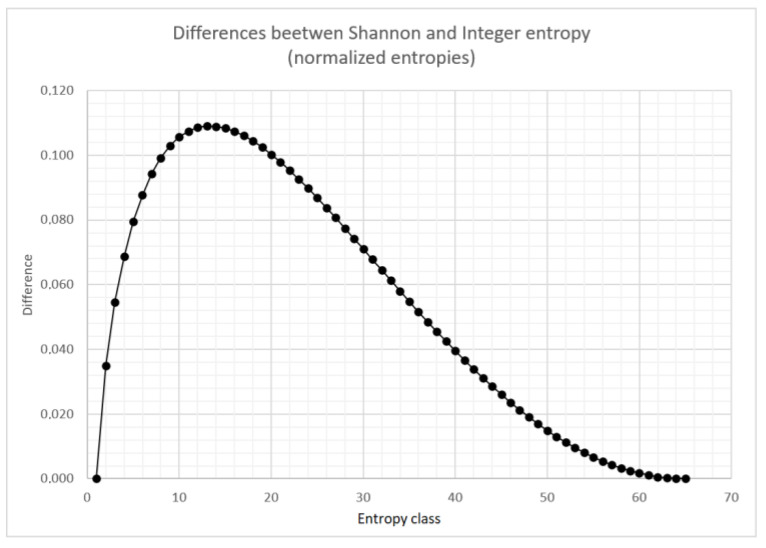
Differences between positional entropy and Shannon entropy for 65 entropy classes.

**Figure 5 entropy-22-01173-f005:**
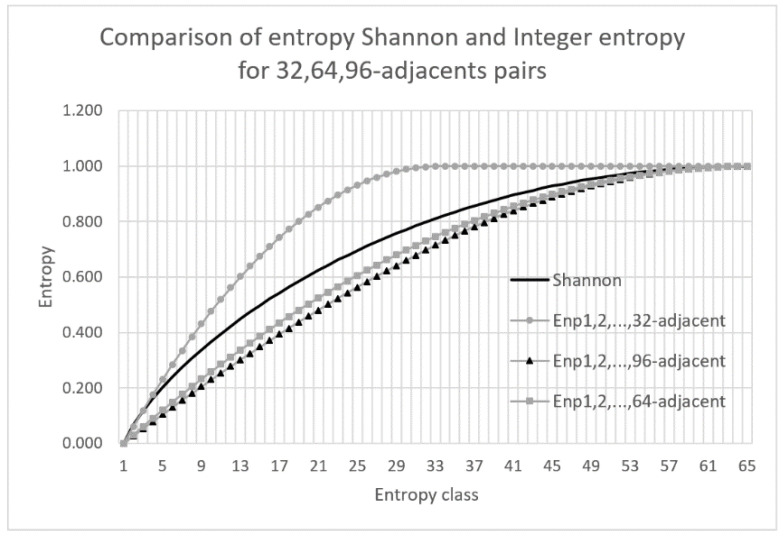
Comparison of the cumulative entropy for 32-, 64- and 96-adjacent pairs with the Shannon entropy.

**Figure 6 entropy-22-01173-f006:**
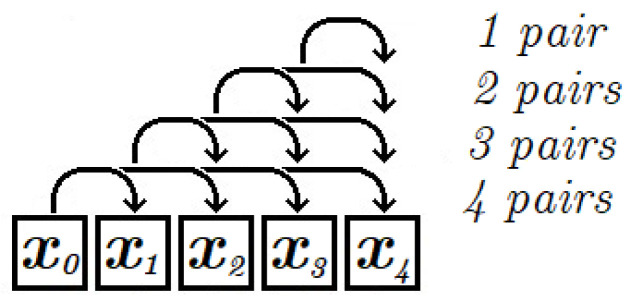
Pairs generated by Algorithm 3 for sequence of 5 elements with *k* = 1.

**Figure 7 entropy-22-01173-f007:**
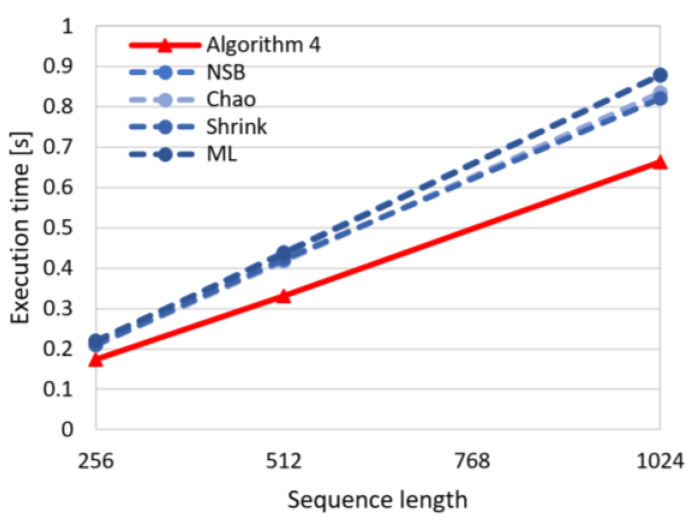
Dependency between execution time and sequence length for the compared algorithms.

**Figure 8 entropy-22-01173-f008:**
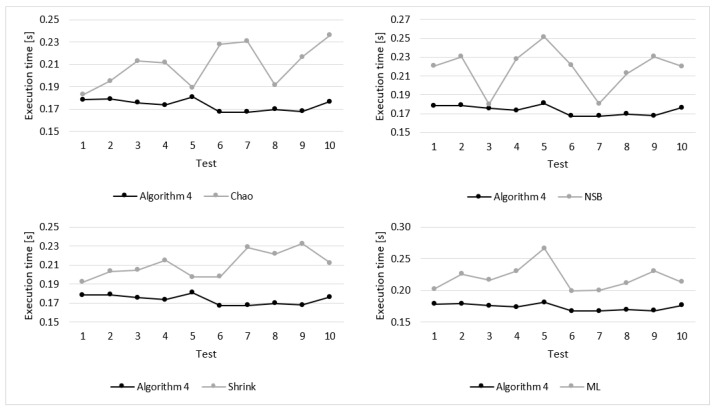
Execution times measured for the compared algorithms during 10 tests (*n* = 256).

**Figure 9 entropy-22-01173-f009:**
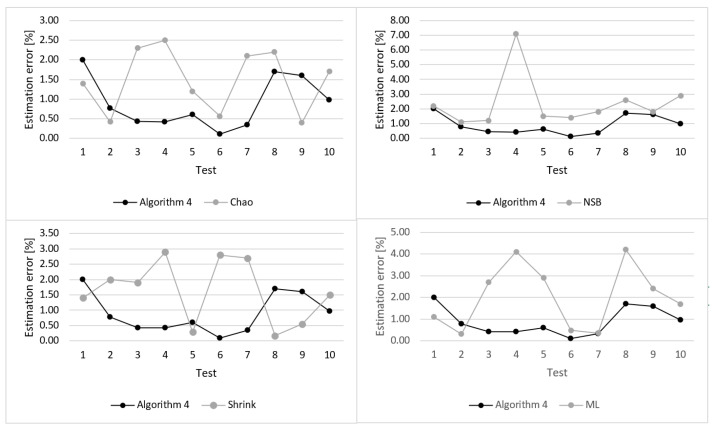
Estimation errors evaluated for the compared algorithms during 10 tests (*n* = 256).

**Table 1 entropy-22-01173-t001:** Comparison of Shannon entropy and positional entropy for 1-adjacent, 2-adjacent and 1,2,3-adjacent pairs.

No.	Sample	Shannon Entropy *Hq*			Positional Entropy 1-Adjacent		Positional Entropy 1,2-Adjacent		Positional Entropy 1,2,3-Adjacent	
		*q* = 2	*q* = *e*	*q* = *10*	$\bar{Enp_{1}(X)}$	*Enp_1_*(X)	$\bar{Enp_{1,2}(X)}$	*Enp_1,2_*(X)	$\bar{Enp_{1,2,3}(X)}$	*Enp_1,2,3_*(X)
1	<0000>	0	0	0	0	0	0	0	0	0
2	<0001>	0.811	0.562	0.244	(1)	(0.334)	(2)	(0.4)	3	0.5
3	<0010>	0.811	0.562	0.244	(2)	(0.667)	(3)	(0.6)	3	0.5
4	<0011>	1	0.693	0.301	1	0.334	3	0.6	4	0.667
5	<0100>	0.811	0.562	0.244	2	0.667	3	0.6	3	0.5
6	<0101>	1	0.693	0.301	(3)	(1)	4	0.8	4	0.667
7	<0110>	1	0.693	0.301	(2)	(0.667)	4	0.8	4	0.667
8	<0111>	0.811	0.562	0.244	1	0.334	2	0.4	3	0.5
9	<1000>	0.811	0.562	0.244	1	0.334	2	0.4	3	0.5
10	<1001>	1	0.693	0.301	(2)	(0.667)	4	0.8	4	0.667
11	<1010>	1	0.693	0.301	(3)	(1)	4	0.8	4	0.667
12	<1011>	0.811	0.562	0.244	2	0.667	3	0.6	3	0.5
13	<1100>	1	0.693	0.301	1	0.334	3	0.6	4	0.667
14	<1101>	0.811	0.562	0.244	(2)	(0.667)	(3)	(0.6)	3	0.5
15	<1110>	0.811	0.562	0.244	(1)	(0.334)	(2)	(0.4)	3	0.5
16	<1111>	0	0	0	0	0	0	0	0	0

**Table 2 entropy-22-01173-t002:** The 1-adjacent positional entropy for different sample types.

Type of Sample *X*	Alphabet Length	Average Enp_1_(*X*)
Binary sequence	2	0.4962
English text	32	0.9593
Digital image-bitmap	256	0.9836
Digital image-compressed	256	0.9964

**Table 3 entropy-22-01173-t003:** Comparison of average execution times and congruity for the proposed algorithms and existing solutions.

Algorithms	Parameters or Methods	Congruity and Execution Time for Sequences of Length *n*					
		*n* = 256		*n* = 512		*n* = 1024	
		Average Execution Time [s]	Congruity [%]	Average Execution Time [s]	Congruity [%]	Average Execution Time [s]	Congruity [%]
Algorithm 2	*m* = *n*/4	0.1814	98.76	0.3519	98.43	0.7098	98.73
Algorithm 3	*k* = 12	0.1798	98.89	0.3587	98.76	0.7183	98.47
Algorithm 4	α = 4	0.1734	99.11	0.3307	99.25	0.6635	99.59
CRAN entropy	Chao	0.2093	98.52	0.4173	98.32	0.8346	98.23
CRAN entropy	NSB	0.2172	97.64	0.4390	98.58	0.8791	98.09
CRAN entropy	Shrink	0.2104	98.37	0.4216	98.69	0.8207	98.34
CRAN entropy	ML	0.2198	98.01	0.4365	98.56	0.8769	98.29

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
