# Peer review of "Application of Positional Entropy to Fast Shannon Entropy Estimation for Samples of Digital Signals"

_entropy, 2020, doi:10.3390/e22101173_

Round 1

Reviewer 1 Report

Thank you for allowing me to review your work.

Although the authors have done interesting work, I do not see a substantial change concerning existing proposals.

In this sense, I would ask the authors for greater clarity on this and in particular, analyzing (do a graph) in detail with the results of Table 3.

Personally, before accepting a work of these characteristics, I would like to observe a substantive change with metrics already used.

Minor details are English correction in various parts of the text
(I am not a native English speaker but I have been able to detect errors in this regard).

Reviewer 2 Report

The paper is devoted to a significant problem for practice: to construct a functional that approximates Shannon entropy estimator but has much smaller computational complexity.  The authors propose the Positional entropy and illustrate their properties by computer experiments.

Unfortunately, the paper contains some inaccuracies:

  1. Theoretical analysis of performance for the proposed entropy estimator is absent.
  2.  In the proof of Theorem 1 it is desirable to explain the case d>1 in more details.
  3.  Line 269. Please explain the parameter α.
  4.  Line 102. Please correct  the formula (4).
  5.  Line 131. Please correct the text.
  6.  Line 166. Please correct the formula for t: t= 1+ n/2.
  7.  Line  167 Correct the formula for t: t=Ceil (n/2).
  8.  Correct misprints on Lines 14, 73, 90.

Reviewer 3 Report

  1.  The results of this paper would appear to be of interest to many researchers who read this journal. The plots, diagrams and Tables appear to be packed with useful information.
  2.  I found the paper very hard to read, perhaps because I only have limited experience with computer science papers.  I suggest some changes to the English usage below in the minor comments further below.
  3. My main concern is that the paper reads like a summary report of what the authors have done, without their providing the means to replicate the results.  Several algorithms are spelled out, and reference to CRAN code, but there are no R scripts that would enable the reader to verify  results of Section 5 or obtain new ones for their own applications. 
  4. I have a problem with Section 2.2.  For p=2 and X={X_1,X_2}, are the X_i 's to be treated as sets when saying their intersection is empty?  Why not just say the p elements of X and are distinct?  
  5. Definition 1 is for 'Real Shannon entropy'.   By "Real" do the authors distinguish this definition form "Complex"? Or do they mean Shannon's original definition is this one given in set notation form. 
  6. Which of the many definitions is introduced elsewhere (give references) and which are new? 
  7. There appears to be three Definition 10's.  Are so many definitions needed?   

Some examples of unclear or wrong English usage:

  • line 28, what is 'element distribution'?
  • lines 43-46:  do you mean 'Such an approach introduces an estimation error; however, it is sufficient for comparison of the entropy between data samples.' 

While the last point appears trivial, when replicated  many times in the paper, (I could go on), it is irritating to the reader and shows the authors haven't asked a native English writer to proofread it.  Also, I am puzzled by the phrase "data sample"; perhaps this is standard usage amongst computer scientists.    

I recommend a major revision with the goal of making the methodology proposed by the authors easy to implement, and a more reader-friendly presentation.  

Round 2

Reviewer 2 Report

In the proof of Theorem 1 it is desirable to analyse the case of even values of d in more details.

Author Response

Reviewer’s comment:

In the proof of Theorem 1 it is desirable to analyse the case of even values of d in more details.

Answer:

The following discussion was added in the revised manuscript (page 8) to explain the case of even d values.

It should be noted here that the binary data samples considered in the above proof have no difference pairs for even values of d. However, the total number of difference pairs is maximal because for odd d values all pairs are the difference pairs. E.g., in case of sample (1, 0, 1, 0, 1, 0) there is no difference pair for d = 2 and d = 4, while there are 5 difference pairs for d = 1, 3 difference pairs for d = 3, and 1 difference pair for d=5.  Thus, the total number of difference pairs reaches the maximum of n2/4 = 9 pairs (n = 6).

Reviewer 3 Report

The authors have substantially improved the readability of

the manuscript.  

Author Response

Thank you very much for the useful comments of our manuscript.